

# Wind turbine impact on operational weather radar I/Q data: characterisation and filtering

Lars Norin[1]

[1]Atmospheric Remote Sensing Unit, Research Department, Swedish Meteorological and Hydrological Institute, Norrköping, Sweden

*Correspondence to:* Lars Norin (lars.norin@smhi.se)

**Abstract.** For the past two decades wind turbines have been growing in number all over the world as a response to the increasing demand for renewable energy. However, the rapid expansion of wind turbines presents a problem for many radar systems, including weather radars. Wind turbines in line-of-sight of a weather radar can have a negative impact on the radar's measurements. As weather radars are important instruments for meteorological offices, finding a way for wind turbines and
weather radars to co-exist would be of great societal value.

Doppler weather radars base their measurements on in-phase and quadrature phase (I/Q) data. In this work a month worth of recordings of high resolution I/Q data from an operational Swedish C-band weather radar are presented. The impact of point targets, such as masts and wind turbines, on the I/Q data is analysed and characterised. It is shown that the impact of point targets on single radar pulses is manifested as a distinct and highly repeatable signature. The shape of this signature is found
to be independent of the size, shape, and yaw angle of the wind turbine. It is further demonstrated how the robustness of the point target signature can be used to identify and filter out the impact of wind turbines in the radar's signal processor.

## 1   Introduction

Wind turbines worldwide are rapidly growing in numbers to meet the increasing demand for renewable energy (Global Wind Energy Council, 2016). Wind farms, densely spaced clusters of wind turbines, are now common sights both on- and offshore.
During the past two decades wind turbines as well as wind farms have been growing in size. A modern wind turbine can reach 200 m in height and wind farms with hundreds of densely spaced turbines have started to appear. As more and more countries set ambitious goals to increase their share of energy from renewable sources, many more wind turbines and wind farms are expected to be built in the future.

The increasing number of wind turbines and wind farms is, however, not without problems. Over the past decades it has be-
come clear that wind turbines can have a negative impact on many electromagnetic communication devices, such as air surveillance radars (Lemmon et al., 2008; Borely, 2010; Theil et al., 2010; Lute and Wieserman, 2011), maritime radars (Howard and Brown, 2004; Rashid and Brown, 2007; Grande et al., 2014), radio links (Bacon, 2002; Randhawa and Rudd, 2009; Lehpamer, 2013), broadcast communication (Sengupta and Senior, 1979; Wright and Eng, 1992; International Telecommunication



Union, 2015), and, the subject of this work, weather radars (Crum et al., 2008; Burgess et al., 2008; Gallardo-Hernando and Pérez-Martínez, 2009; Norin and Haase, 2012).

Weather radars are important instruments for meteorological offices. Data from weather radars are, for example, used by meteorologists to follow the weather in real time, as input to numerical weather prediction models (e.g. Ridal and Dahlbom, 2016), and to drive hydrological models (e.g. Berg et al., 2016). Wind turbines in line-of-sight of a weather radar can have a negative impact on the radar's measurements. The rotating blades of wind turbines defeat conventional ground clutter filters, commonly installed in Doppler weather radars. The failure to suppress echoes from the wind turbines' rotating blades leads to erroneous estimations of the radar moments, i.e. the radar reflectivity factor (hereafter referred to as reflectivity), radial velocity, and spectrum width. These errors can thereafter propagate to affect derived radar products, such as precipitation rate. Finding a way for wind turbines to co-exist with weather radars would therefore be of great societal value.

In order to better understand the impact of wind turbines on weather radars several studies have been dedicated to characterise wind turbine-contaminated radar products. Burgess et al. (2008), Crum and Ciardi (2010), and Vogt et al. (2011) presented data from US S-band weather radars to demonstrate the impact of wind turbines on the radar products while Haase et al. (2010), Norin and Haase (2012), and Norin (2015a) used long time series of operational weather radar products from Swedish C-band radars to study the same phenomenon. They all found that wind turbines can have a profound impact on all of the radar products and that the impact of wind turbines is highly variable with time.

To mitigate the impact of wind turbines on weather radar products various methods have been suggested. Isom et al. (2009) proposed multi-quadratic interpolation over contaminated areas using data from neighbouring, non-contaminated radar cells while Aarholt and Jackson (2010) and Norin (2015b) suggested the use of gap-filling radars to replace measurements corrupted by wind turbines. These mitigation techniques are promising but interpolating radar products may lead to unnecessary large losses of information and while gap-filling radars can replace contaminated measurements they are likely a costly solution.

Another possible way to reduce the impact of wind turbines on weather radars is to develop filters, acting on low level data in the radars' signal processors. Doppler radar products are based on in-phase and quadrature phase (I/Q) data, which are usually collected with much higher resolution than the radar products. Filtering the impact of wind turbines in the radar I/Q data may therefore lead to a smaller loss of information compared to interpolating the radar products.

Efforts have been made to analyse and characterise the impact of wind turbines on weather radar I/Q data. Gallardo et al. (2008) recorded I/Q data from a Spanish C-band weather radar while Isom et al. (2009) collected I/Q data from two US S-band weather radars. By directing the radar towards the wind turbines they found that the turbines give rise to complex but characteristic patterns in the time-frequency domain. Mitigation schemes, based on the observed spectral characteristics of wind turbines, have nonetheless been proposed (Gallardo et al., 2008; Bachmann et al., 2010a, b).

However, to observe the characteristic pattern of a wind turbine in the frequency domain the radar should dwell on the target for a period of time. For operational weather radars, using a scanning mode to measure the surrounding atmosphere, there is no time to do so. Methods for scanning weather radars have therefore also been suggested. Gallardo-Hernando et al. (2010) proposed a method to identify wind turbines, based on zero-Doppler echo power and spectrum width, while Nai et al. (2011) suggested using signal processing in the range-Doppler domain to detect and remove the impact of wind turbines. While these





suggestions are promising it remains difficult to correctly identify and separate wind turbine-contaminated spectra from spectra containing precipitation echoes.

In this work we present wind turbine-contaminated I/Q data, recorded by a recently modernised Swedish C-band weather radar during one month of normal operation. Taking advantage of the high sampling rate of the radar the I/Q data were examined in the time domain instead of the frequency domain. It is shown that point targets, such as wind turbines and masts, have a similar, characteristic signature, easily recognisable in echoes from single radar pulses. The point target signature is shown to be very robust and is independent of wind turbine size, shape, and yaw angle. To demonstrate that this feature can be exploited by mitigation techniques a simple filter, capable of identifying and removing wind turbine impact directly in the radar's signal processor, is presented.

The paper is organised in the following way. The Swedish weather radars are described in Sect. 2 and the recorded sets of I/Q data, together with all other data sources, are presented and discussed in Sect. 3. Results of the I/Q data analyses are presented in Sect. 4 while the wind turbine filter is described in Sect. 5. The paper is summarised and concluded in Sect. 6.

## 2 The Swedish weather radars

The Swedish weather radar network consists of 12 C-band Doppler weather radars. The weather radars scan the surrounding atmosphere by continuously rotating around their own axis using different elevation angles. A full set of scans, with different elevation angles, is referred to as a polar volume.

The oldest weather radars in the Swedish network use single, horizontal polarisation measurements and date from the early 1990ies. In September 2014 a modernisation process started, upgrading the Swedish weather radars from single to dual polarisation. By the end of 2016 four of the radars had been upgraded. The modernisation process is expected to be completed in 2018. More details on the old radars are given by, e.g., Michelson (2006); Norin (2015a).

In addition to utilising two polarisations the modernised Swedish radars also provide a much higher sampling rate compared to the older radar systems. For the upgraded radar systems a range resolution of 15.625 m is possible, compared to 167 m for the older systems. The new radars also provide user-adjustable settings for most of the radar parameters such as pulse repetition frequency (PRF), rotational speed, the range- and azimuth resolution of the radar products as well as the possibility to select between four different pulse lengths. Relevant parameters for the new radar systems are presented in Table 1.

In contrast to the old radars systems the modern Swedish weather radars offer the possibility to record and store I/Q data. Each modern radar is equipped with a 5 TB hard drive where I/Q recordings can be stored for further analysis. Furthermore, with the new radars it is also possible to add custom filters to the radars' signal processors, making it possible to, e.g., design and implement wind turbine filters acting directly on I/Q data.

Polar volumes from the modernised weather radars are completed every 5 min. Conveniently, there are some extra time slots available between some of the scans during which additional measurements can be performed. For the purpose of this study, one of these extra time slots was used for the recording of I/Q data.





## 3 Data sets

In order to study the impact of wind turbines on weather radar I/Q data one of the modernised Swedish weather radars, Vara (56.2859° N, 12.8120° E), was selected. Radar Vara is well suited for this purpose since 45 wind turbines are located in line-of-sight of the radar within a radius of 15 km. The modernised radar Vara has been operational since May 2016, delivering
radar products to the Swedish Meteorological and Hydrological Institute (SMHI).

During September 2016 complete scans of I/Q data were recorded for the lowest elevation angle (0.5°), once every hour. The data collection started on 8 September at 06 UTC and was temporarily stopped on Friday 9 September at 09 UTC. On Monday, 12 September the recording was resumed at 07 UTC and continued uninterrupted until 29 September, 13 UTC. Every recorded scan was made up by approximately 10500 pulses and every pulse was sampled 6400 times in range. In total 447 complete
scans were recorded during this period. In order to study the effect of different radar pulse lengths 18 additional complete scans were recorded during 11 October 2016. During these scans the pulse length was varied between the four selectable options: 0.5 μs, 0.8 μs, 1.0 μs, and 2.0 μs. Relevant radar parameters used during the I/Q recordings are listed in Table 1.

During September 2016 the weather in the region near radar Vara was mostly clear except during 26–28 September when bands of rain passed over the radar from the south and the west, providing an opportunity to study the impact of wind turbines
during precipitation. During the measurements in October the weather was clear.

In order to analyse the effect of wind turbine yaw angle (which depends on the wind direction) the local wind speed and wind direction were obtained from one of SMHI's automatic weather stations (58.3221° N, 13.0406° E), located approximately 14 km to the northeast of radar Vara. Once every hour this station reports the average wind direction and wind speed measured during 10 min. The measurements are made 10 m above the ground.
Finally, a flight obstacle database, issued by LFV (the Swedish civil aviation authority) was used to find the positions of wind turbines and masts located near radar Vara. The positions of all obstacles used in the study were confirmed visually using satellite images.

## 4 Results and discussion

Weather radar Vara is surrounded by wind turbines and masts. Within a radius of 15 km there are 45 wind turbines and 30
masts listed in the flight obstacle database. The impact of these point targets can be seen clearly in Fig. 1 which shows the average, total (unfiltered) reflectivity product measured by radar Vara during September 2016. From Fig. 1 it can be seen that, in general, average reflectivity values ranged from below 10 dBZ up to 35 dBZ. However, in several locations small areas with reflectivities larger than 50 dBZ can be seen. Many of these locations correspond to the locations of known point targets, such as wind turbines and masts. Other obstacles in line-of-sight of the radar, such as tall buildings (silos, church towers)
located in the nearby town Vara situated approximately 10 km–15 km to the east of the radar, also give rise to large average reflectivity values. From Fig. 1 it is clear that the radar reflectivity product and its derivatives (e.g. precipitation rate) would benefit from reducing the impact from these point targets. The application of a conventional clutter filter would suppress echoes from stationary targets, such as masts, but echoes from moving targets, such as wind turbines, would still remain.





In order to investigate the similarities and differences between the impact of stationary and moving point targets on weather radar I/Q data the echoes from a nearby mast are examined in Sect. 4.1 while the echoes from a nearby wind turbine are analysed in Sect. 4.2. In these sections it is shown that these point targets give rise to a characteristic, repeatable signature in the I/Q data that is easily recognisable in echoes from single radar pulses. In Sect. 4.3 these results are generalised by examining the

signatures of 20 different wind turbines, with varying sizes and shapes. The robustness of the point target signature is further investigated by examining echoes from wind turbines with different yaw angles as well as by changing the radar pulse lengths.

## 4.1   Impact of a mast on weather radar I/Q data

Before examining the impact of wind turbines we start by analysing echoes from a mast, located 4 km to the west of radar Vara. The mast is in line-of-sight of the radar and extends well into the half-power beamwidth of the radar main lobe for the

lowest elevation angle, assuming standard atmospheric propagation conditions.

Figure 2 shows the impact of the mast on the radar I/Q data. The average amplitude of all valid I/Q scans, recorded during September 2016, from an area around the mast is shown in Fig. 2a. A distinct increase in amplitude, reaching a maximum around 300 m behind the obstacle, can be seen. Figure 2b shows the average phase gradient for the same area. It is clear that in regions where the corresponding amplitude is increased, the phase gradient is near zero.

Profiles of the amplitude and phase gradient as functions of azimuth, at the distance where the mean amplitude reached its maximum (around 300 m behind the mast), are shown in Fig. 2c and Fig. 2d, respectively. Figure 2c reveals that as the radar beam moves near the mast the amplitude increases smoothly until the radar beam is centred on the obstacle. After this point the amplitude decreases, symmetrically to the increase. Variations from the median amplitude, here represented by the 5th and 95th percentiles, are seen to be very small. The amplitude profile from a single scan is also shown in Fig. 2c. While not as

smooth as the median amplitude, the difference in shape is minimal.

The median phase gradient profile, shown in Fig. 3d, is close to zero for all azimuth angles. For azimuth angles where the amplitude is increased the variations of the phase gradient profiles are small. However, for the azimuth angles where the corresponding amplitude is small, i.e. where there is no influence from the mast, the variations in the phase gradient are much larger. For completeness the phase gradient profile from a single scan is also shown.

Figure 2e and Fig. 2f show profiles of amplitude and phase gradient as functions of distance, for the azimuth angle where the mean amplitude reached its maximum (see Fig. 3a). A prominent shape of the amplitude profile, resembling the absolute value of the sinc function, is seen in Fig. 2e. This amplitude profile is highly repeatable, as revealed by the small variations. It is interesting that while the transmitted radar pulse is close to rectangular the recorded echoes from the mast have a different shape. Most likely, this is due to changes in echo power that can occur when the signal is passed through the radar receiver

(see, e.g. Doviak and Zrnić, 2006, p. 74).

The profile of the corresponding phase gradient, shown in Fig. 2f, reveals that the median phase gradient is near zero at distances where the amplitude is increased. The variations in the phase gradient are also seen to be small at these distances. However, at distances where the corresponding amplitude values are small, the variations in the phase gradient profiles are large.



The small variations in the amplitude and phase gradient profiles in the area influenced by the mast reflect the fact that the echoes from the stationary mast are very robust and do not change much between different scans.

## 4.2 Impact of a wind turbine on weather radar I/Q data

To complement the results for the stationary mast presented above let us now examine the impact of a moving point target, a wind turbine.

Figure 3 shows the impact of a wind turbine, located 3 km to the northeast of radar Vara. The average amplitude of the I/Q data from September 2016 is shown in Fig. 3a. As for the mast, a distinct increase in amplitude can be seen, reaching a maximum around 300 m behind the turbine. Figure 3b shows the corresponding phase gradient. As for the mast, in regions where the amplitude is increased the phase gradient is near zero. On average, the impacts of the mast and of the wind turbine are, as expected, very similar. To see the difference in impact between the stationary mast and the moving wind turbine data from individual scans must be examined.

Profiles of the amplitude and phase gradient as functions of azimuth, at the distance where the mean amplitude reached its maximum value, are shown in Fig. 3c and Fig. 3d. Figure 3c shows that the median amplitude profile increases smoothly until the radar beam is centred on the wind turbine. After this point the amplitude decreases. In contrast to the mast, the wind turbine amplitude profiles show large variability and the amplitude values from single scans do not follow a smooth curve. As an example Fig. 2c also shows an amplitude profile from a single scan. The amplitude values are seen to vary sharply between neighbouring pulses. This is the reason for the complex spectral patterns of wind turbines observed in the frequency domain. The 5th and 95th percentiles of all amplitude profiles show that amplitude variations are large, ranging from 25 % to over 200 % of the median amplitude values. This reflects the fact that during the period the data were collected the wind turbine blades were in different positions, giving rise to very different values of the radar cross section. The median phase gradient profile, shown in Fig. 3d, is close to zero for all azimuth angles. As for the mast, for azimuth angles where the corresponding amplitude values are increased the phase gradient variations are small, albeit slightly larger than for the mast (cf. Fig. 2d). For the azimuth angles where the corresponding amplitude values are small the variations in the phase gradient are larger. A phase gradient profile from a single scan is also shown, confirming these observations.

Figure 3e and Fig. 3f show profiles of amplitude and phase gradient as functions of distance, for the azimuth angle in which the maximum average amplitude of the wind turbine is observed (cf. Fig. 3a). The shape of the median amplitude profile in Fig. 3e is very similar to the median profile of the mast (cf. Fig. 2e). Even though the shape of the amplitude profiles are very robust, the maximum amplitude value of the different profiles show large variations. An amplitude profile from a single scan reveals that the characteristic shape is intact, even for echoes from a single pulse. The corresponding phase gradient, shown in Fig. 3f, reveal that the phase gradient is near zero when the amplitude is increased. Again, at distances where the corresponding amplitude values are increased the variations in the phase gradient profiles are small. However, at distances where the amplitude values are small, the variations of the phase gradient are large. The phase gradient profile from a single scan is also shown, for completeness.





The profiles of the amplitude and phase gradient as functions of distance, i.e. for individual pulses, are remarkably similar for the wind turbine and the mast. The main difference is that the amplitude values between neighbouring pulses vary much more for the wind turbines than for masts. This is a consequence of the wind turbine's rotating blades and is also the reason for the complex spectral patterns that wind turbines generate, which has been reported in many studies (see, e.g., Poupart, 2003; Gallardo et al., 2008; Isom et al., 2009). However, for single pulses it seems that the shapes of the amplitude and phase gradient profiles are robust, regardless if the target is moving or not. It therefore appears that point targets can be easily recognised in echoes from single radar pulses. In the following text these profile shapes are referred to as the point target signature.

## 4.3 Robustness of the point target signature

In order to generalise the results of the point target signature, presented in Sect. 4.1 and Sect. 4.2, the robustness of the signature needs to be examined further. First, the signature of a single wind turbine is examined for a large number of pulses with varying echo strength. Next, the signatures from 20 different wind turbines, all located near radar Vara, are analysed. Finally, the significance of wind turbine yaw angle is investigated as well as the consequences of changing the radar pulse length.

First, let us examine the normalised amplitudes and the absolute values of the phase gradient in echoes from the wind turbine that was analysed in Sect. 4.2 (located 3 km to the northeast of radar Vara). From Fig. 3a it is obvious that the wind turbine affects radar measurements within an azimuth range of at least $\pm 0.5°$ from its position, due to width of the radar main lobe (cf. Table 1). Pulses within $\pm 0.5°$ azimuth of the wind turbine were therefore extracted from all scans of I/Q data that were recorded during September 2016. In total 12459 valid pulses were found. In order to compare the amplitude shape from pulses with different echo strengths the data were normalised. The amplitude values from every pulse were normalised by the pulse's amplitude value at the distance where the mean amplitude profile reached its maximum (around 300 m behind the turbine, cf. Fig. 3a).

The median, normalised amplitude profile from these pulses, together with profiles representing the 5th and 95th percentiles, are shown in Fig. 4a. The characteristic shape of the point target signature can be seen, having very small variations in amplitude values. Only far from the wind turbine where the normalised amplitude values were weak, below 0.15 in this case, can the 5th and the 95th percentiles be seen to differ more than approximately 3 % from the median values. The reason for this deviation is that the weaker the echo strength of the wind turbine, the greater the influence from other echoes (e.g. from noise or precipitation), which most likely do not have the same signature as the point target.

Figure 4b shows the 5th, 50th, and 95th percentiles of the corresponding absolute phase gradient profiles. Again, the point target signature is clearly recognised from previously presented results (cf. Fig. 3d). Variations from the median, absolute phase gradient values are small near distances where the amplitude is increased, and otherwise larger.

In Fig. 3 it was shown that the echo strength from a wind turbine can vary sharply from pulse to pulse. However, from the results in Fig. 4 it can be concluded that the characteristic shape of the wind turbine signature is unaffected by strength of the maximum amplitude value. The point target signature can therefore be recognised in the echoes from any single radar pulse as long as these echoes are stronger than simultaneous echoes from precipitation or noise.





To investigate whether the point target signature varies for different wind turbine models or wind turbine sizes, I/Q data from 20 wind turbines located in line-of-sight of weather radar Vara within a distance of 15 km, were analysed. Amplitude and phase gradient data were extracted from pulses directed towards the selected wind turbines from all scans of I/Q data that were recorded by radar Vara during September 2016. Figure 5 shows the average, normalised amplitude and the average, absolute

phase gradient of the 20 wind turbines. From Fig. 5a it can be seen that the mean amplitude profiles for the different wind turbines are almost identical. Differences between the profiles can only be found far from the turbines, where the amplitude values are small. These differences are most likely the effect of noise or precipitation, which occasionally overpowers some of the echoes from the wind turbines.

The average, absolute phase gradient profiles for all 20 wind turbines are shown in Fig. 5b. Again, a striking similarity

between the profiles can be observed for distances where the corresponding amplitude values are increased. Further away from the maximum amplitude value (greater than $\pm 300$ m) the absolute phase gradient profiles from the different wind turbines are seen to vary much more. This is expected as the impact of the wind turbines is very small there.

From the close similarities in the signatures of all 20 wind turbines it is clear that these obstacles, as well as the mast described in Sect. 4.1 above, have a highly repeatable and distinct signature in the radar I/Q data. The size and shape of a wind

turbine seem to have a negligible impact on the point target signature.

Depending on wind direction, wind turbines rotate around their yaw axis to maximise their harvest of wind energy. As wind turbines with different yaw angles lead to very different values of the radar cross section, as seen by the weather radar, it is important to examine whether the wind turbine yaw angle affects the point target signature.

Using data from an automatic weather station, located approximately 14 km to the northeast of radar Vara (see also Sect. 3),

the prevailing wind direction and wind speed for all scans recorded during September 2016 were extracted. The most common wind direction during this period was from the southwest. All scans with a wind direction between 230° and 240° and with a wind speed greater than 3 m s$^{-1}$, were therefore selected. To examine the influence of the wind turbine yaw angle the amplitude and phase gradient from four different wind turbines were analysed. Two of the examined wind turbines were located to the northeast of the radar (bearing 48.4° and 50.7°), i.e. almost parallel to the wind direction. During the selected scans their yaw

angles were therefore close to 5°, as seen by the weather radar. The other two wind turbines were located to the northwest (bearing 345.4° and 345.5°), i.e. perpendicular to the wind direction, with yaw angles near 90° from the weather radar. As the radar cross section values for wind turbines with such yaw angles are very different (Angulo et al., 2015) the impact on the wind turbine signature, if any, should be made visible by this analysis.

The normalised amplitude profiles from the selected scans for all four wind turbines are shown in Fig. 6a. It can be seen that

all amplitude profiles are very similar. No difference between the profiles can be discerned except far from the turbines, where the amplitude values are very low. As discussed above, this is the result of echoes from noise or precipitation. Correspondingly, the absolute phase gradient profiles, shown in Fig. 6b, are also very similar. Deviations in the absolute phase gradient can only be seen far from the amplitude maximum, where the influence of the wind turbines is negligible. From the results in Fig. 6 it can therefore be concluded that wind turbine yaw angle does not seem to have any significant impact on the point target

signature.





So far it has been shown that the point target signature is remarkably robust, unaffected by echo strength as well as by wind turbine shape, size, and yaw angle. However, changing the radar pulse length could, and should, change the signature of a point target. To examine this effect 18 additional scans of I/Q data were recorded on 10 October 2016 with varying radar pulse lengths. For the modernised Swedish radars it is possible to select between four different pulse lengths: 0.5 μs, 0.8 μs, 1.0 μs,

and 2.0 μs (cf. Table 1). To investigate the impact of pulse length the amplitude and phase gradient data from pulses directed towards the mast analysed in Sect. 4.1 were extracted.

Figure 7a shows the amplitude profiles for the different pulse lengths. It is seen that even though the shape of the amplitude profiles resemble each other for pulse lengths up to 1.0 μs, the shape is more divergent for the 2.0 μs pulse length. This could be due to difference in the shapes of the transmitted pulse or of a different behaviour of the radar receiver or both. This is not

possible to determine without a thorough investigation and is out of scope of this paper. The corresponding absolute phase gradient is shown in Fig. 7b. Again, it can be observed that the phase gradient profiles for pulse lengths up to 1.0 μs are similar while the phase gradient profile of the 2.0 μs pulse is more different.

Even though the point target signatures clearly change with changing pulse lengths, very small variation in the respective signatures were observed for the analysed data. This implies that the point target signatures are robust, albeit slightly different,

after changing the pulse length. It is clear that the point target signature must be determined specifically for the parameters used by the radar. It should also be emphasised that these results are based on the I/Q recordings from the Swedish weather radar Vara. Even though point targets from other weather radars also should have distinct and robust signatures, their exact shapes may differ from the results presented here.

## 5   Filtering the impact of wind turbines

In Sect. 4.3 it was shown that point targets, such as masts and wind turbines, exhibit a robust and characteristic signature that can be recognised in the echoes from single radar pulses. In this section a simple filter, capable of suppressing the impact of wind turbines during clear weather as well as during precipitation, is described.

The filter consists of two parts, identification and cleaning. For the filter to be useful for an operational weather radar the identification and cleaning of point target signatures should be automatic and reliable. After identifying a point target signature

the cleaning part of the filter should attempt to recreate the conditions of the surrounding, unaffected echoes, whether from clear air targets or precipitation. Both the identification and the cleaning algorithm must take into the account the restrictions of the operational weather radar's signal processor. After describing the filter its efficiency is tested by applying it to the recorded I/Q data.

### 5.1   Automatic identification of wind turbines

The first step to successfully filter the impact of wind turbines in weather radar I/Q data is to automatically identify their signature, which ideally should be possible during conditions of clear weather as well as during precipitation. One way to





identify wind turbine echoes in I/Q data from a single radar pulse is to match the amplitude and phase gradient data to the known, ideal point target signature presented in Sect. 4.

The ideal point target signature for radar Vara, when using a pulse length of 0.5 μs, is here defined as the median, normalised amplitude profile and median, absolute phase gradient profile shown in Fig. 4. The ideal signature was limited to a distance of ±523 m from the distance of the maximum amplitude. This corresponds to a window length of 67 samples. This ideal signature was used to find matching data points in the I/Q data.

Even though the signature of a point target was shown in Sect. 4 above to be very distinct and robust it is necessary to first investigate whether a similar signature can also occur from precipitation. To examine the uniqueness of the point target signature the signature was matched to full scans of I/Q data. To find a match the ideal signature was shifted through the samples from every pulse in a scan. To find matching data points the amplitude values of the I/Q data were normalised by the amplitude value of the centre point in the 67 samples long window. A data point was considered a match if the absolute difference between the normalised amplitude of the examined samples and the amplitude of the ideal signature was less than 3 % and the absolute difference between the absolute phase gradient and the ideal absolute phase gradient was less than 3 rad. km$^{-1}$ (cf. the variations seen in Fig. 4). The results showed that near known wind turbines the number of matching data points were higher than elsewhere. It was also observed that the number of matching data points in general were higher during clear conditions compared to when precipitation was present. This can be explained by noticing that more point targets are normally visible when there are no echoes from precipitation present to mask their signatures.

In Fig. 8 an example of identifying point target signatures in I/Q data from a single radar pulse is presented. Figure 8a and Fig. 8b show amplitude and phase gradient data near a wind turbine together with the ideal point target signature. Data points matching the ideal signature are highlighted. Figure 8c shows the number of data points that were found to match the ideal point target signature, when shifting the 67 samples long window sample by sample from 0 km to 15 km. It is seen that the number of matching data points are different during clear conditions and during precipitation. During precipitation the only locations where the number of matching data points were higher than 15 correspond to the locations of known wind turbines. During clear conditions, a few more locations with more than 15 matching points were found. As mentioned above, this is due to more, albeit weak, point targets are visible during clear conditions.

To claim identification a point target a minimum number of matching data points must be chosen. To ensure a high probability of detection, requiring fewer matching data points would be better. However, fewer matching data points also leads to a higher false alarm rate. For this work, a threshold of 13 data points were chosen which corresponds to the number of data points in the ideal signature whose amplitude values are higher than 0.5. This means that wind turbine echoes exceeding underlying reflectivity data by up to 3 dB will not be identified but at the same time limiting the false alarm rate. In order to reduce the false alarm rate even further the identification algorithm was only applied in the vicinity of known wind turbines. Matching was made to pulses within ±2.5° in azimuth and between -125 m and 500 m in range from the known location of a target.

It is worth pointing out that when implementing a similar detection algorithm for an operational weather radar, it may a good idea to search for the wind turbine signature within some extra degrees in azimuth. This is because unless the radar has a





continuous calibration of bearing it may easily drift a little which could throw the identification off, if it is only applied at the locations of known wind turbines.

## 5.2  Suppressing the impact of wind turbines

After identifying wind turbine-contaminated data points, the impact of the wind turbines should be cleaned. Even though

identification of the point target signature is made in echoes from single radar pulses it is advantageous to clean contaminated data using two-dimensional interpolation, i.e. to also use data from surrounding, uncontaminated pulses. The reason for using a two-dimensional interpolation technique is that sharp changes in amplitude and phase between neighbouring pulses can otherwise be introduced, which can lead to errors in radial velocity or spectrum width.

For this simple filter the identified wind turbine-contaminated data were cleaned using natural neighbour interpolation (Sib-

son, 1981; The MathWorks, 2016). Natural neighbour interpolation can be used on data sets where scattered data points are missing and it provides a smooth approximation of the surrounding, uncontaminated data. An example of identification and cleaning of wind turbine-contaminated data is presented in Fig. 9. The original amplitude and phase data are shown in Fig. 9a and Fig. 9c, respectively. The data points for which the point target signature have been identified are highlighted. Figure 9b and Fig. 9d show the amplitude and phase data after cleaning. It is seen that the large increase in amplitude, caused by the wind

turbine, has been removed and that the phase now changes smoothly from pulse to pulse.

From Fig. 9 it can also be noted that pulses containing identified point target signatures are found connected within an area almost 2°wide in azimuth. However, for a wind turbine-filter to be useful for an operational weather radar the radar's signal processing technique must be taken into account. Radar Vara calculates radar moments using pulses collected within 1°azimuth (cf. Table 1). Hence, it may not always be possible for the cleaning algorithm to rely on surrounding, uncontaminated data.

Nevertheless, data can still be cleaned using natural neighbour interpolation. To see the efficiency of the filter, adjusted for radar Vara's operational settings, the identification and cleaning algorithms were applied to two full scans of I/Q data. After first subjecting pulses within ±2.5°of a known wind turbine to the filter reflectivity values were recreated from I/Q data, as done by radar Vara's signal processor.

Figure 10 shows two examples of recreated reflectivity data near radar Vara. Figure 10a shows recreated reflectivity from

the original I/Q data during clear conditions on 9 September 2016 at 09 UTC without applying the wind turbine filter. After applying the identification algorithm and cleaning the identified data points using natural nearest neighbour interpolation, the wind turbine-filtered, recreated reflectivities are shown in Fig. 10b. It can be seen that much of the wind turbine-contaminated data are suppressed or removed completely. In some locations there are two or more wind turbines located closely in range, affecting the same radar pulses. The impact of these targets superpose, which this simple filter was not designed to handle. In

these locations the identification algorithm struggles and some of the wind turbine-contaminated data remain.

Figure 10c shows unfiltered, recreated reflectivity data from the original I/Q measurements from 28 September 2016 at 04 UTC when bands of rain passed over radar Vara from the east. After applying the identification and cleaning algorithms to the I/Q data the resulting, filtered reflectivities are shown in Fig. 10d. The filter is seen to accurately remove the wind turbine





impact while leaving the precipitation echoes intact. However, as before, when multiple targets are closely spaced radially from the radar the filter fails to identify and remove their superposed signatures.

Nonetheless, it can be concluded that by applying this simple filter to I/Q data from radar Vara a large improvement in the weather radar's reflectivity data can be achieved.

## 6  Summary and conclusions

During the past two decades wind turbines have increased rapidly in numbers all over the world as a response to the increasing demand for renewable energy. However, it has also become clear that radar systems, such as weather radars, are easily disturbed by wind turbines located in line-of-sight of the radar. As weather radars are important tools for meteorological offices, finding a way for wind turbines and weather radars to co-exist would be of great societal value.

One way to reduce the impact of wind turbines on weather radar measurements would be to install a filter in the radar's signal processor acting directly on low level radar measurements, the in-phase and quadrature phase (I/Q) data. Doppler weather radars use I/Q data to estimate radar moments (such as reflectivity, radial velocity, and spectrum width) in the frequency domain. However, research has shown that separating the impact of wind turbines from precipitation echoes in the frequency domain is very difficult due to the highly complex and time-varying spectral patters that are generated by the turbines.

In this work the impact of wind turbines on weather radar in-phase and quadrature phase (I/Q) data has been examined. I/Q data from a Swedish C-band weather radar, capable of sampling data every 15.625 m, were recorded during September and October 2016. Taking advantage of the high sampling rate of the radar, data were analysed in the time domain. By examining the impact of a stationary point target (mast) and moving point targets (wind turbines), similarities and differences in the recorded I/Q data were revealed. It was shown that the echo strength from a stationary point target change smoothly from pulse
to pulse while for the wind turbines, amplitude values can vary sharply between neighbouring pulses. This amplitude variation is the reason for the complex patterns wind turbines exhibit in the frequency domain and is also why their impact is difficult to mitigate.

By analysing the amplitude and phase of point target-impacted I/Q data it was shown that both stationary and moving point targets have a characteristic and highly repeatable signature. Even though amplitudes values in echoes from wind turbines
can vary sharply from pulse to pulse the signature shape was shown to be remarkably robust. Furthermore, the point target signature was shown to be independent of wind turbine size, model, and yaw angle. However, the exact shape of the point target signature was shown to be a property of the settings of the radar. Changing the radar pulse length was shown to alter the point target signature.

The distinct and repeatable signature of point targets can be used to identify and remove the impact of wind turbines. A
filter, capable of identifying and removing echoes contaminated by wind turbines, was presented. The design of this filter was simple and robust, so as to be possible to implement in an operational weather radar's signal processor. By applying the filter to full scans of recorded I/Q data and recreating the radar reflectivity product, it was shown that the impact of wind turbines can be significantly suppressed, both during clear weather as well as during precipitation.





Even though it is clear that wind turbine impact on the radar reflectivity product can be removed or suppressed the possibility to mitigate the impact of wind turbines on other radar moments, such as radial velocity and spectrum width, should be investigated further. To test the filter additionally, more I/Q data should be recorded, preferably during a variety of meteorological conditions. A working filter should also be implemented into a radar's signal processor and tested during operational

5   conditions. It would also be valuable if high resolution I/Q data from a different radar could be recorded and compared to the presented results.

## 7   Data availability

Radar products (such as reflectivity) or derived product (such as precipitation rate) for research purposes are available on request from SMHI. Data from Swedish automatic weather stations (such as wind speed and wind direction, used in this work)

10   are available from `opendata-catalog.smhi.se/explore/`

*Competing interests.* The author declares that he has no conflict of interest.

*Acknowledgements.* This work was financed by the Swedish Energy Agency as a part of the feasibility study Wind turbine filters for weather radars.



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





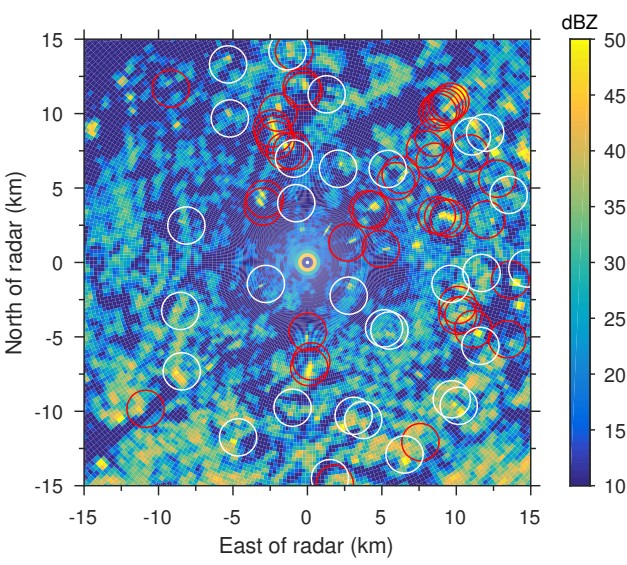

**Figure 1.** Average, total (unfiltered) reflectivity data from radar Vara for September 2016. Red circles are centred on locations of known, nearby wind turbines while white circles mark the positions of known masts.







**Figure 2.** Impact of a mast on weather radar I/Q data, recorded by radar Vara during September 2016. Panel **(a)** shows the average I/Q amplitude near the position of the mast, while panel **(b)** shows the average phase gradient. The position of the mast is marked by a black or white circle. Black and white vertical and horizontal lines indicate from where data in panels **(c)** to **(f)** originate. Panels **(c)** and **(d)** show the 5th, 50th and 95th percentiles of the amplitude and phase gradient, respectively, as functions of azimuth. Panels **(e)** and **(f)** show the 5th, 50th and 95th percentiles of the amplitude and phase gradient, respectively, as functions of distance. Black lines in panels **(c)** to **(f)** show example data from a single scan.





**Figure 3.** Impact of a wind turbine on weather radar I/Q data, recorded by radar Vara during September 2016. Panel **(a)** shows the average I/Q amplitude near the position of the wind turbine, while panel **(b)** shows the average phase gradient. The position of the wind turbine is marked by a white circle. Black and white vertical and horizontal lines indicate from where data in panels **(c)** to **(f)** originate. Panels **(c)** and **(d)** show the 5th, 50th and 95th percentiles of the amplitude and phase gradient, respectively, as functions of azimuth. Panels **(e)** and **(f)** show the 5th, 50th and 95th percentiles of the amplitude and phase gradient, respectively, as functions of distance. Black lines in panels **(c)** to **(f)** show example data from a single scan.





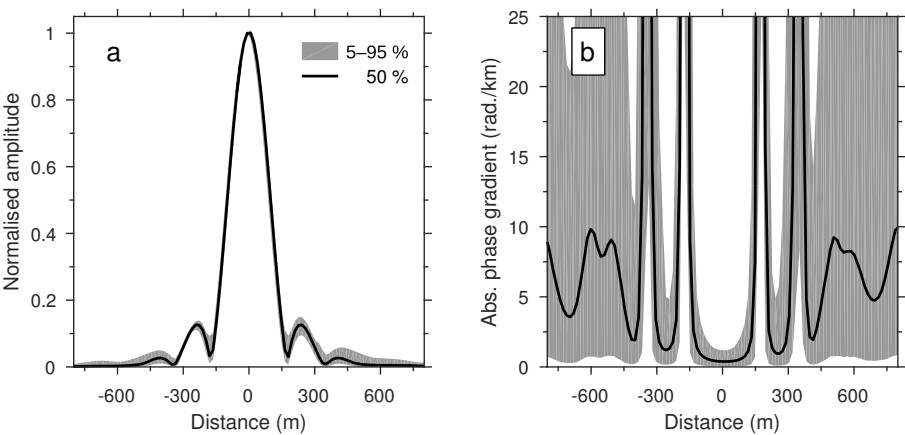

**Figure 4.** Impact of a single wind turbine on weather radar I/Q data, recorded by radar Vara during September 2016. Panel **(a)** shows the median, normalised amplitude together with the 5th and 95th percentiles. Panel **(b)** shows the corresponding absolute phase gradient.





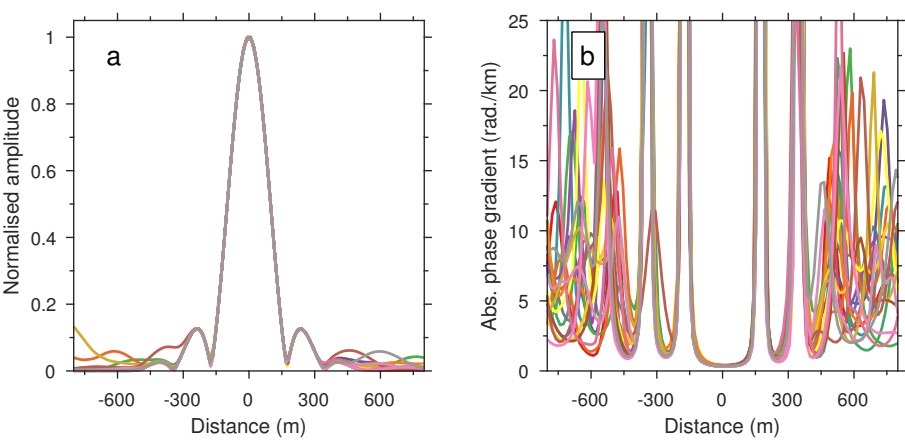

**Figure 5.** Impact of 20 wind turbines on weather radar I/Q data, recorded by radar Vara during September 2016. Panel **(a)** shows the average, normalised amplitudes of the wind turbine echoes while panel **(b)** shows the corresponding average, absolute phase gradient values.




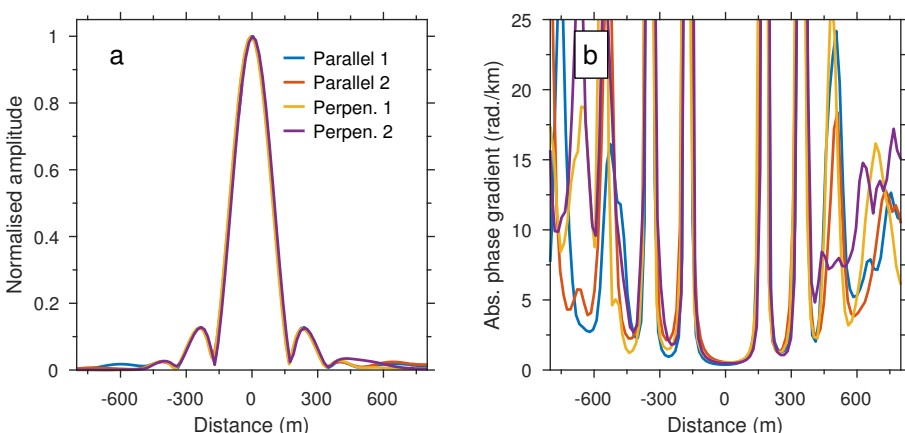

**Figure 6.** Impact of four wind turbines on weather radar I/Q data, recorded by radar Vara during conditions with wind direction from the southwest. Two turbines were located northeast of the radar, parallel to the wind direction. The other two turbines were located to the northwest, perpendicular to the wind direction. Panel **(a)** shows the average, normalised amplitudes and panel **(b)** shows the corresponding average, absolute phase gradients.




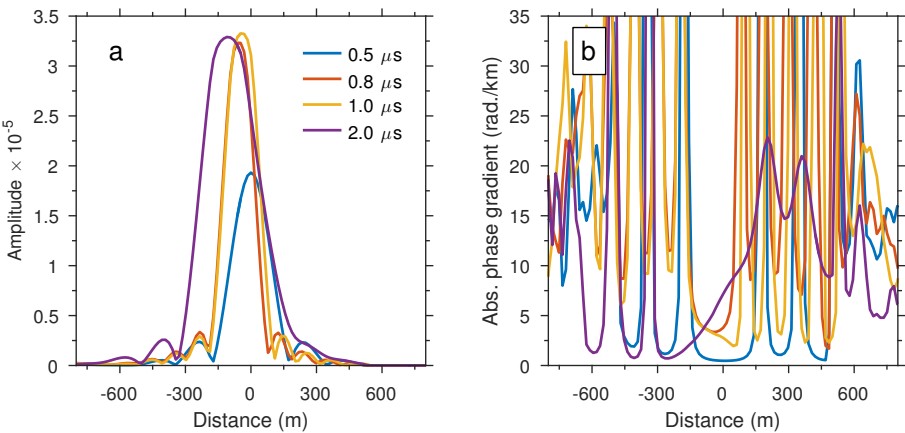

**Figure 7.** Impact of a nearby mast on weather radar I/Q data using four different pulse lengths. The data were recorded by radar Vara on 11 October 2016. Panel **(a)** shows the average, normalised amplitudes and panel **(b)** shows the corresponding average, absolute phase gradients.



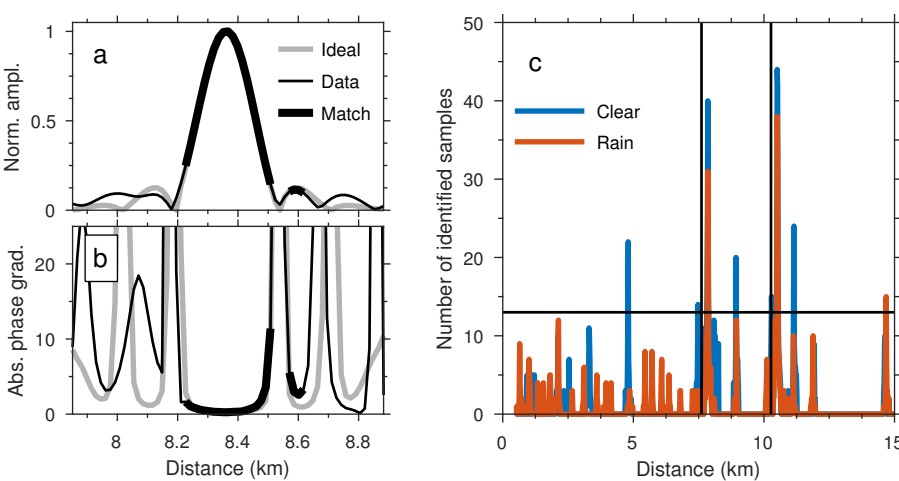

**Figure 8.** Matching data points from a single radar pulse to the ideal point target signature. Panel **(a)** shows normalised amplitude data, the corresponding ideal point target signature, and matching data points in the vicinity of a wind turbine. Panel **(b)** shows the absolute phase gradient, the corresponding ideal point target signature, and matching data points. Panel **(c)** shows the number of matching data points when shifting the ideal point target signature along data from a single radar pulse during two occasions: one during clear weather, the other during precipitation. Vertical lines show the location of two wind turbines and the horizontal line shows the minimum number of data points used in this work to identify a point target.





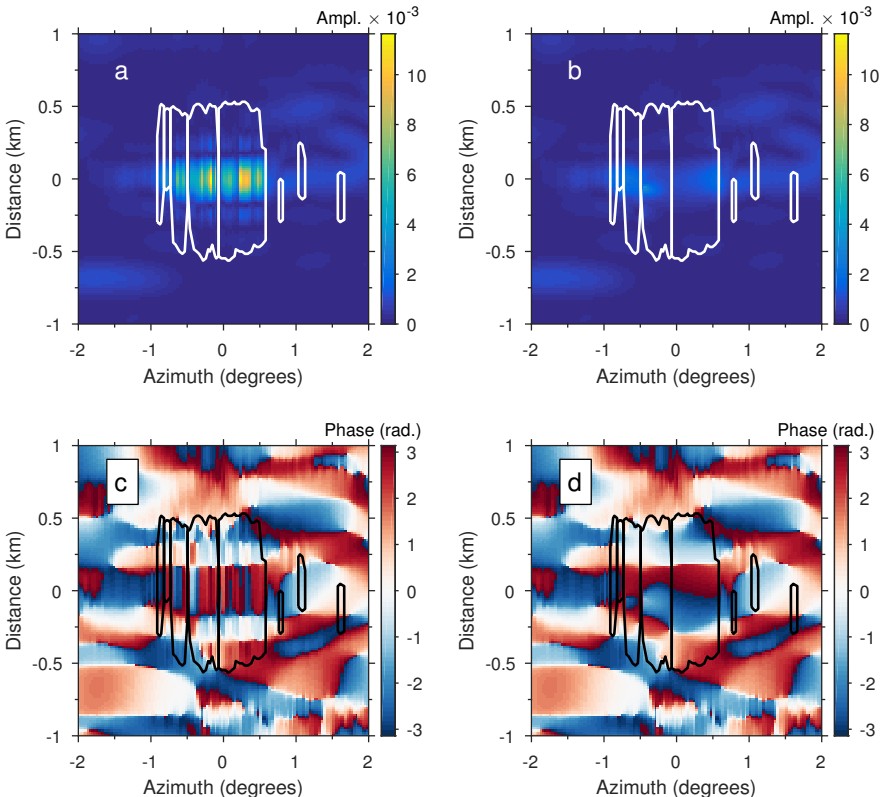

**Figure 9.** Identifying and filtering the impact of a wind turbine on weather radar I/Q data recorded on 8 September 2016, at 07 UTC. Panel (a) shows the amplitude of the recorded I/Q data near the wind turbine. Data found to match the point target signature are enclosed by white lines. Panel (b) shows the amplitude of the filtered I/Q data. Panel (c) shows the phase of the recorded I/Q data with the wind turbine-impacted areas encircled in black. Panel (d) shows the phase of the filtered data.



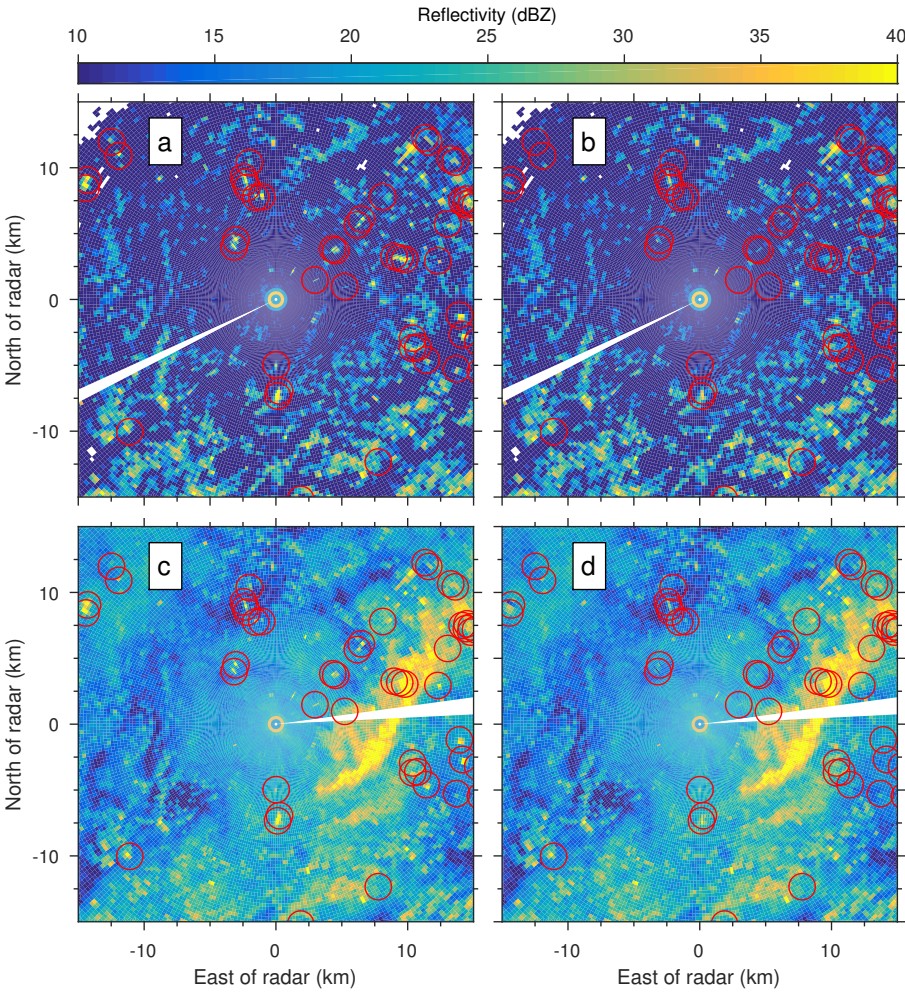

**Figure 10.** Filtering the impact of wind turbines. Top row shows recreated reflectivity data from 9 September 2016 at 09 UTC. No precipitation occurred near radar Vara at this time. Panel **(a)** shows recreated reflectivities from the original I/Q data, with the positions of known wind turbine marked by red circles. Panel **(b)** shows recreated reflectivity data after applying a wind turbine filter to the I/Q data. The bottom row shows recreated reflectivity data from 28 September 2016 at 04 UTC. During this time rain bands passed over radar Vara from the west. Panel **(c)** shows recreated reflectivity from the original I/Q data while panel **(d)** shows filtered, recreated reflectivities.

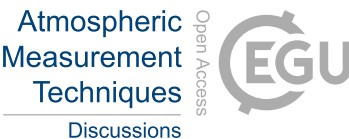

**Table 1.** Parameters used by weather radar Vara during the recording of I/Q data in September and October 2016.

| | |
|---|---|
| Elevation angle | 0.5° |
| Transmit power | 250 kW |
| Gain | 45 dB |
| Beamwidth | 1.0° |
| Wavelength | 5.35 cm |
| Rotational speed | 3 rpm |
| PRFs | 600/450 Hz |
| Maximum range | 100 km |
| Range resolution (I/Q data) | 15.625 m |
| Range resolution (products) | 250 m |
| Azimuthal resolution (products) | 1.0° |
| Pulse width (September) | 0.5 µs |
| Pulse width (October) | 0.5 µs, 0.8 µs, 1.0 µs, 2.0 µs |