# Peer review of "Wind turbine impact on operational weather radar I/Q data: characterisation and filtering"

_Atmospheric Measurement Techniques, 2017_

## Referee Comment (RC1) · D. de la Vega (Referee) · 20 Feb 2017

General comments

The paper addresses the problem caused by the clutter from wind turbines on weather radars. It characterise the clutter signature, in order to automatically identify echoes from wind turbines. Additionally, it proposes a filtering method to identify and remove the clutter from wind turbines that provides quite good results. Both the characterization of the wind turbine signatures and the evaluation of the efficiency of the proposed filtering technique are based on a huge dataset recorded by a weather radar of the SMHI network.

Though previous works have been developed in the recent years to characterize the signature echoes from wind turbines, all of them were developed under the assumption

of a fixed radar beam pointing at the wind turbine during several seconds. Till now, none of these studies has analysed the real situation of a scanning radar beam, where the wind turbine is illuminated during a very short time. In this last case, the signature from the wind turbine is more difficult to obtain, and therefore, to filter it out in the radar receiver. This study addresses this complex situation, based on a wide dataset from measurements and a thorough analysis of the recorded data. Additionally, a filtering technique is proposed and applied to the recorded dataset. Results seem to be really good and the proposed filtering technique seems to be efficient. Though there is still further investigation to be done (close wind turbines and effect on other radar moments such as radial speed and spectrum width), the research line is really promising and encouraging.

Specific comments

The paper is adequate for the topics and the scope of AMT. The topic is timely and of interest for the research community. It addresses a significant problem for weather agencies that requires detailed research. The manuscript is well structured and the organization of the contents is logical and easy to follow. The abstract provides a clear and complete summary of the content of the paper. The methodology, the results and the subsequent analysis are well argued and described. The description of the analysis is sufficiently detailed to allow the reproduction of a similar study. The figures are clear and they provide helpful information for a better understanding The conclusions are thoroughly argued and discussed along the text of the paper; they are in line with the results shown in the paper. The methodology is properly referenced; references included in the manuscript are appropriate and provide a good view of the state of the art of the recent research. In summary, the quality of the paper is very high, the methodology is solid and the results useful and interesting for the research community.

Questions to the author

Author states that the shape of the normalised signature is independent of the yaw

angle of the wind turbine. Nevertheless, results from several studies referenced in the manuscript show differences in the amplitude of the echoes up to 40-50 dB, due to the variability of the scattering pattern of the rotor. One of the references of the paper demonstrates that these variations do not occur for a short range of elevation angles, as the mast is the main reflector in this sector (Angulo et al., 2015, referenced in page 8, line 27). Does the author think that this is the reason of the absence of variability with the yaw angle? If not, what could be the reasons that could justify this unexpected result?

Technical corrections

Just a minor comment about the description of the results. In some sentences, mainly those in the first part of the manuscript, where the methodology is still to be described, it should be clearly specified if signatures from wind turbines are normalised or not; otherwise, this may cause ambiguity. As an example, the sentence in the abstract "...is manifested as a distinct and highly repeatable signature. The shape of this signature is found to be independent of the size, ..." may be understood as the size of the turbine is not relevant for a not-normalised signature. A review according this aspect is recommended.

---

## Referee Comment (RC2) · J. Kurdzo (Referee) · 20 Mar 2017

General Comments

The author presents an analysis in the time/range domain of I/Q signals near and around masts and wind turbines using a C-band weather radar. Many of the pre-existing studies look at different domains, meaning that this manuscript is a useful contribution to the literature. Of particular note is the presentation of exceptionally robust signal characteristics, especially with changes in blade yaw. The phase gradient plots are particularly interesting.

The manuscript is well-written, has a logical flow, and maintains excellent graphical presentation. The graphics are clear and well-labeled and will allow a casual reader to quickly identify the usefulness of the manuscript. I have some concerns about citations

and the general claims of performance that I have addressed in my Major Comments, and I have listed several Specific Comments for the author's attention. I know that I have made a lot of comments, but please don't be discouraged; I think the manuscript will be more than acceptable for AMT after a round of revisions, and I fully support its eventual publication. Great job!

Major Comments

1. In general, too many of your citations are from conference presentations. Many of these presentations have resulted in journal publications afterwards, and those are much more appropriate to cite. This is especially relevant for the AMS conferences, as those papers and abstracts are not heavily peer reviewed. There are also areas where inappropriate references were made; I have tried to mention some of these areas.

2. The main issue I have with the manuscript is that it isn't made clear until the very end that the filtering is only applied near "known" locations of wind turbines. I may be wrong, but don't many other cited works in your manuscript assume that the locations are not known? This is important for several reasons. Of course, new wind turbines may be installed; wind turbines may be in an "on" or "off" state, changing their signature; and different atmospheric conditions will change the propagation characteristics of the beam, meaning not only could the signatures change, but they may be non-existent during cases of sub-refraction, or more prominent during super-refraction or ducting, for example. The distinction of whether or not you know where the targets are is important, and can be misleading to the reader. For example, on Page 3 / Line 8, you say that the feature can be exploited to "identify" and remove the turbine. But really the signature is just being used to remove the turbine where you expect it to already be, correct? This doesn't change the usefulness of the manuscript, but I think you should set up the reader in such a way that this will only be applied in the appropriate places. It may be appropriate both in the abstract and the introduction.

3. The removal of contaminated data in precipitating echoes is not quantified. We don't

see "before" and "after" spectra in precipitation; this is something I would expect to see. We also don't have an easy truth to compare with, meaning there would need to be a way to quantify the added and removed bias. This is all ok, but it means that using words such as "significantly" improved data is inappropriate. If the improvement is only shown qualitatively (i.e., before and after pictures of the data) rather than quantitatively, the claims must be quelled to what the data actually show. If we don't have clues regarding the quantitative decrease in bias in precipitating echoes, claims regarding "significant" improvements should not be made.

Specific Comments

1. Page 2, Line 4: I think there are better references you can cite here; the Ridal and Dahlbom paper is still in review, and there are older references you can cite. Here are some examples I would recommend instead:

Sun, J. and J. Wilson, 2003: The assimilation of radar data for weather prediction. Meteorological Monographs, 52, 175–198.

Xue, M., D. Wang, J. Gao, K. Brewster, and K. K. Droegemeier, 2003: The Advanced Regional Prediction System (ARPS), storm-scale numerical weather prediction and data assimilation. Meteorology and Atmospheric Physics, 82, 139-170.

Zhao, Q., J. Cook, Q. Xu, and P. Harasti, 2006: Using radar wind observations to improve mesoscale numerical weather prediction. Wea. Forecasting, 21, 502–522.

2. Page 2, Line 5: Similar to previous comment; the Berg et al. citation isn't really appropriate here. Cite the seminal papers. Here are some examples:

Corral, C., D. Sempere-Torres, M. Revilla, and M. Berenguer, 2000: A semi-distributed hydrological model using rainfall estimates by radar. Application to Mediterranean basins. Physics and Chemistry of the Earth, Part B: Hydrology, Oceans and Atmosphere, 25, 1133-1136.

Carpenter, T. M., K. P. Georgakakos, and J. A. Sperfslagea, 2001: On the parametric

and NEXRAD-radar sensitivities of a distributed hydrologic model suitable for operational use. Journal of Hydrology, 253, 169-193.

Ganguly, A. and R. Bras, 2003: Distributed quantitative precipitation forecasting using information from radar and numerical weather prediction models. J. Hydrometeor., 4, 1168–1180.

Gourley, J. and B. Vieux, 2005: A method for evaluating the accuracy of quantitative precipitation estimates from a hydrologic modeling perspective. J. Hydrometeor., 6, 115–133.

3. Page 2, Line 6: I would make it clearer why a conventional ground clutter filter doesn't work here. Mention that a notch filter at zero Doppler doesn't work due to the blades being in near-constant motion during any existence of wind.

4. Page 2, Line 11: Add a comma between "weather radars" and "several studies"

5. Page 2, Line 13: Change "US" to "United States" (this applies across the manuscript, e.g., Line 27 on the same page). Another option is to use "U.S." (with periods).

6. Page 2, Lines 20-21: The final sentence of this paragraph is a run-on sentence. Break it up with appropriate commas.

7. Page 2, Lines 23-24: It is misleading to say that I/Q data are collected at much higher resolution. What about a short-pulse radar that doesn't use any range oversampling? The sampling rate could easily be identical to the range resolution of the radar. If you mean that there are more samples in time (which I wouldn't really consider temporal resolution, per se), say that instead. Changing this sentence may require changes to the following sentence.

8. Page 2, Line 34: The Nai et al. reference is from an older conference paper. I would cite their most recent journal paper as follows instead:

Nai, F., S. Torres, and R. Palmer, 2013: On the mitigation of wind turbine clutter for

weather radars using range-Doppler spectral processing. IET Radar, Sonar & Navigation, 7, 178-190.

9. Page 3, Line 4: Add a comma between "of the radar" and "the I/Q data"

10. Page 3, Line 18: Change "1990ies" to "1990s"

11. Page 3, Line 21: Add a comma between "two polarisations" and "the modernised"

12. Page 2, Line 22: No change necessary here, but note that your higher sampling rate is not a universal standard for weather radars (see back to Specific Comment 7). So maybe back on Page 2 you can specify that this is the case for the Swedish radars, specifically.

13. Page 3, Line 26: "radars" should be "radar" and a comma is needed after "radar systems"

14. Page 4, Line 9: Rather than provide the number of pulses by "scan," it would be preferable to mention roughly how many pulses are used for a given azimuth, for example. In other words, how many pulses did you use to process the individual gates that were contaminated? The number of pulses in the scan is irrelevant in my opinion.

15. Page 4, Line 17: Define the SMHI acronym for the unfamiliar reader.

16. Page 4, Line 20: Same as above for the LFV acronym.

17. Page 5, Lines 1-3: This is a run-on sentence. Break it up with commas in the appropriate locations.

18. Page 5, Line 13: Can you explain why the maximum amplitude is so far "behind" the mast location rather than at the mast location? Maybe I missed something.

19. Page 6, Lines 7-8: It sounds awkward to start back-to-back sentences with "As for the mast. . ."

20. Page 7, Lines 6-7: It should be made clear that, so far, you have only indicated

that these targets can be easily recognized without the presence of weather or other echoes.

21. Page 7, Line 16: Change "due to width of the radar main lobe" to "due to the width of the radar main lobe"

22. Page 9, Lines 7-8: Can you comment on why the "centers" of the amplitudes change by nearly 150 m? Is this a sampling issue? I wouldn't think so with a sampling rate of ∼15 m.

23. Page 9, Line 10: Change "is out of the scope of this paper" to "is beyond the scope of this paper"

24. Page 9, Lines 13-18: I think it is quite a stretch to say that the point target signatures are "robust, albeit slightly different" when changing the pulse length. The difference between pulse lengths is significantly higher (Figure 7) than in your previous comparisons. Maybe just acknowledge this difficulty and move on; but I don't think they are similar enough to use the same descriptors from earlier sections.

25. Page 10, Lines 33-34: Change "it may a good idea" to "it may be a good idea"

26. Page 11, Lines 9-15: Was this case during precipitation? If so, it should be noted.

27. Page 11, Lines 21-23: This is a run-on sentence. Add commas where appropriate.

28. Page 12, Lines 3-4: Regarding the final sentence in this section; the improvement of the reflectivity factor data is qualitative at best; no bias estimation is provided, especially in the precipitation regions. You have suppressed turbines in some instances, but it's not appropriate to conclude that you have made a "large improvement" on the data. The subjectivness of this means the sentence should be removed.

29. Page 12, Line 24: "amplitudes" should be singular

30. Page 12, Line 33: Remove "significantly" – it is subjective, as we have seen nothing quantitative to suggest a statistically significant removal of clutter, especially in precip-

itation. The results are simply plots of before and after; they are qualitative in nature.

---

## Author Comment (AC1) · 10 Apr 2017

I thank the referee for the time and effort devoted to review this manuscript as well as for the constructive comments and suggestions together with the encouraging words. Below, please find a point-by-point reply to the comments (reproduced in italics).

*Questions to the author*

*Author states that the shape of the normalised signature is independent of the yaw angle of the wind turbine. Nevertheless, results from several studies referenced in the manuscript show differences in the amplitude of the echoes up to 40–50 dB, due to the variability of the scattering pattern of the rotor. One of the references of the paper demonstrates that these variations do not occur for a short range of elevation angles, as the mast is the main reflector in this sector (Angulo et al., 2015, referenced in page*

*8, line 27). Does the author think that this is the reason of the absence of variability with the yaw angle? If not, what could be the reasons that could justify this unexpected result?*

I agree that the amplitudes of wind turbine echoes can vary greatly depending on rotor blade orientation and/or the yaw angle. It is also true that as the influence of the wind turbine tower increases, the variation in the echo amplitudes decreases. However, from Fig. 3c it can be seen that a large variation in amplitude exists, suggesting that the echo amplitudes are not dominated by the tower.

The absence of variation in the wind turbine signatures for different yaw angles can be explained by the fact that the signatures are normalised by the maximum amplitude value. They are therefore independent of the echo strength. For a single pulse, the wind turbine appears like a point target. Once normalised by the maximum amplitude, the signature becomes independent of the rotor blade orientation and the yaw angle.

*Technical corrections*

*Just a minor comment about the description of the results. In some sentences, mainly those in the first part of the manuscript, where the methodology is still to be described, it should be clearly specified if signatures from wind turbines are normalised or not; otherwise, this may cause ambiguity. As an example, the sentence in the abstract "...is manifested as a distinct and highly repeatable signature. The shape of this signature is found to be independent of the size, ..." may be understood as the size of the turbine is not relevant for a not-normalised signature. A review according this aspect is recommended.*

The text in the manuscript has been revised accordingly.
* * *

---

## Author Comment (AC2) · 10 Apr 2017

I thank the referee for his highly detailed review of the manuscript and for the very constructive comments and suggestions. Below, please find a point-by-point reply to the comments (reproduced in italics).

*Major Comments*

*1. In general, too many of your citations are from conference presentations. Many of these presentations have resulted in journal publications afterwards, and those are much more appropriate to cite. This is especially relevant for the AMS conferences, as those papers and abstracts are not heavily peer reviewed. There are also areas where inappropriate references were made; I have tried to mention some of these areas*

[Figure]

Thanks for the suggested references. I have followed your advice where applicable. However, to the best of my knowledge, there are not that many peer-reviewed papers published in the specific topic of wind turbines and weather radars. I have therefore had limited success in finding replacements for several of the conference publications.

*2. The main issue I have with the manuscript is that it isn't made clear until the very end that the filtering is only applied near "known" locations of wind turbines. I may be wrong, but don't many other cited works in your manuscript assume that the locations are not known? This is important for several reasons. Of course, new wind turbines may be installed; wind turbines may be in an "on" or "off" state, changing their signature; and different atmospheric conditions will change the propagation characteristics of the beam, meaning not only could the signatures change, but they may be non-existent during cases of sub-refraction, or more prominent during super-refraction or ducting, for example. The distinction of whether or not you know where the targets are is important, and can be misleading to the reader. For example, on Page 3 / Line 8, you say that the feature can be exploited to "identify" and remove the turbine. But really the signature is just being used to remove the turbine where you expect it to already be, correct? This doesn't change the usefulness of the manuscript, but I think you should set up the reader in such a way that this will only be applied in the appropriate places. It may be appropriate both in the abstract and the introduction.*

One of the major points of this work was to develop an algorithm that is capable of identifying wind turbine echoes for all the conditions you mention (newly installed or unknown turbines, turbines in an "on" or "off" state, turbines changing their signatures due to rotor blade orientation and/or yaw angle, different atmospheric conditions, etc.). Furthermore, the algorithm was also intended to be able to differentiate between echoes from wind turbines and precipitation (as well as wind turbine echoes partially masked by precipitation). One of the main results of this work is that the presented algorithm can do all this. Obviously, I have not succeeded in conveying this message.

It is correct that when creating Fig. 10 the filter was only applied in the vicinity of the locations of known wind turbines. However, this is not a necessary condition for the identification algorithm to work, it was used as a way to lower the false alarm rate.

The algorithm works by identifying point targets in single pulses. As can be seen in Fig. 8c, the algorithm can be applied at any distance in a single pulse (i.e. not only near known wind turbines) and still find the intended targets. However, as the algorithm searches for the signature it normalises the data and may therefore also identify very weak point targets, which may not be of interest. (During precipitation the weak point targets are partially or completely masked which is why during such conditions fewer data points are identified, as seen in Fig. 8c). By providing the algorithm the locations of known wind turbines the false alarm rate can be lowered. For this reason, when creating Fig. 10, the algorithm was applied with the restriction that it should only search near locations of known wind turbines.

While I therefore respectfully disagree with the reviewer on the applicability of the algorithm, I can see that this should have been explained more clearly in the manuscript. I have therefore adjusted the text in several places to make this message come across clearer.

*3. The removal of contaminated data in precipitating echoes is not quantified. We don't see "before" and "after" spectra in precipitation; this is something I would expect to see. We also don't have an easy truth to compare with, meaning there would need to be a way to quantify the added and removed bias. This is all ok, but it means that using words such as "significantly" improved data is inappropriate. If the improvement is only shown qualitatively (i.e., before and after pictures of the data) rather than quantitatively, the claims must be quelled to what the data actually show. If we don't have clues regarding the quantitative decrease in bias in precipitating echoes, claims regarding "significant" improvements should not be made.*

[Figure]

I agree with the reviewer. The text in the manuscript has been revised accordingly.

*Specific Comments*

*1. Page 2, Line 4: I think there are better references you can cite here; the Ridal and Dahlbom paper is still in review, and there are older references you can cite. Here are some examples I would recommend instead:*

*Sun, J. and J. Wilson, 2003: The assimilation of radar data for weather prediction. Meteorological Monographs, 52, 175–198.*

*Xue, M., D. Wang, J. Gao, K. Brewster, and K. K. Droegemeier, 2003: The Advanced Regional Prediction System (ARPS), storm-scale numerical weather prediction and data assimilation. Meteorology and Atmospheric Physics, 82, 139–170.*

*Zhao, Q., J. Cook, Q. Xu, and P. Harasti, 2006: Using radar wind observations to improve mesoscale numerical weather prediction. Wea. Forecasting, 21, 502—522.*

The suggested references have been added to the manuscript.

*2. Page 2, Line 5: Similar to previous comment; the Berg et al. citation isn't really appropriate here. Cite the seminal papers. Here are some examples:*

*Corral, C., D. Sempere-Torres, M. Revilla, and M. Berenguer, 2000: A semi-distributed hydrological model using rainfall estimates by radar. Application to Mediterranean basins. Physics and Chemistry of the Earth, Part B: Hydrology, Oceans and Atmosphere, 25, 1133–1136.*

*Carpenter, T. M., K. P. Georgakakos, and J. A. Sperfslagea, 2001: On the parametric and NEXRAD-radar sensitivities of a distributed hydrologic model suitable for operational use. Journal of Hydrology, 253, 169–193.*

*Ganguly, A. and R. Bras, 2003: Distributed quantitative precipitation forecasting using*

[Figure]

*information from radar and numerical weather prediction models. J. Hydrometeor., 4, 1168—1180.*

*Gourley, J. and B. Vieux, 2005: A method for evaluating the accuracy of quantitative precipitation estimates from a hydrologic modeling perspective. J. Hydrometeor., 6, 115—133.*

The suggested references have been added to the manuscript.

*3. Page 2, Line 6: I would make it clearer why a conventional ground clutter filter doesn't work here. Mention that a notch filter at zero Doppler doesn't work due to the blades being in near-constant motion during any existence of wind.*

Done.

*4. Page 2, Line 11: Add a comma between "weather radars" and "several studies"*

Done.

*5. Page 2, Line 13: Change "US" to "United States" (this applies across the manuscript, e.g., Line 27 on the same page). Another option is to use "U.S." (with periods).*

Done.

*6. Page 2, Lines 20–21: The final sentence of this paragraph is a run-on sentence. Break it up with appropriate commas.*

Done.

*7. Page 2, Lines 23–24: It is misleading to say that I/Q data are collected at much*

*higher resolution. What about a short-pulse radar that doesn't use any range oversampling? The sampling rate could easily be identical to the range resolution of the radar. If you mean that there are more samples in time (which I wouldn't really consider temporal resolution, per se), say that instead. Changing this sentence may require changes to the following sentence.*

The sentence has been rewritten.

*8. Page 2, Line 34: The Nai et al. reference is from an older conference paper. I would cite their most recent journal paper as follows instead:*

*Nai, F., S. Torres, and R. Palmer, 2013: On the mitigation of wind turbine clutter for weather radars using range-Doppler spectral processing. IET Radar, Sonar & Navigation, 7, 178–190.*

Done.

*9. Page 3, Line 4: Add a comma between "of the radar" and "the I/Q data"*

Done.

*10. Page 3, Line 18: Change "1990ies" to "1990s"*

Done.

*11. Page 3, Line 21: Add a comma between "two polarisations" and "the modernised"*

Done.

*12. Page 2, Line 22: No change necessary here, but note that your higher sampling rate is not a universal standard for weather radars (see back to Specific Comment 7). So maybe back on Page 2 you can specify that this is the case for the Swedish radars, specifically.*

Done.

*13. Page 3, Line 26: "radars" should be "radar" and a comma is needed after "radar systems"*

Done.

*14. Page 4, Line 9: Rather than provide the number of pulses by "scan," it would be preferable to mention roughly how many pulses are used for a given azimuth, for example. In other words, how many pulses did you use to process the individual gates that were contaminated? The number of pulses in the scan is irrelevant in my opinion.*

The number of pulses per scan is given to indicate the amount of data processed. The number of pulses per azimuth gate is user-defined. In the operational settings at the time of data recording, an azimuth gate consisted of all pulses collected within one degree. Information on how many pulses are collected within one degree has been added to the manuscript.

*15. Page 4, Line 17: Define the SMHI acronym for the unfamiliar reader.*

SMHI has already been defined on the same page (page 4, line 5).

*16. Page 4, Line 20: Same as above for the LFV acronym.*

LFV was previously an acronym for "Luftfartsverket" but since 2007 they use their

(previous) acronym as their name. Nevertheless, I have changed "LFV" to "Luft-fartsverket/LFV" in the manuscript.

*17. Page 5, Lines 1–3: This is a run-on sentence. Break it up with commas in the appropriate locations.*

Done.

*18. Page 5, Line 13: Can you explain why the maximum amplitude is so far "behind" the mast location rather than at the mast location? Maybe I missed something.*

The range range is calibrated manually for every new radar during the installation. Evidently, the calibration of the range range was not perfect for radar Vara during the time of the data recording. An sentence explaining this has been added to the manuscript.

*19. Page 6, Lines 7–8: It sounds awkward to start back-to-back sentences with "As for the mast..."*

The sentence has been rewritten.

*20. Page 7, Lines 6–7: It should be made clear that, so far, you have only indicated that these targets can be easily recognized without the presence of weather or other echoes.*

Done.

*21. Page 7, Line 16: Change "due to width of the radar main lobe" to "due to the width of the radar main lobe"*

Done.

*22. Page 9, Lines 7–8: Can you comment on why the "centers" of the amplitudes change by nearly 150 m? Is this a sampling issue? I wouldn't think so with a sampling rate of ~15 m.*

The radar range is calibrated manually during the radar installation (see answer to specific comment 18). Furthermore, the radar range is calibrated individually for the different pulse lengths and, evidently, this calibration was not perfect for radar Vara. A sentence explaining this has been added to the manuscript.

*23. Page 9, Line 10: Change "is out of the scope of this paper" to "is beyond the scope of this paper"*

Done.

*24. Page 9, Lines 13–18: I think it is quite a stretch to say that the point target signatures are "robust, albeit slightly different" when changing the pulse length. The difference between pulse lengths is significantly higher (Figure 7) than in your previous comparisons. Maybe just acknowledge this difficulty and move on; but I don't think they are similar enough to use the same descriptors from earlier sections.*

I agree and have rewritten this paragraph. What I meant to describe was that signatures from different pulses with different pulse lengths (e.g. 2.0 $\mu$s) all look equally similar *to each other* as pulses using pulse length 0.5 $\mu$s do. Obviously this was not explained in a clear way. The text in the manuscript has been revised.

*25. Page 10, Lines 33–34: Change "it may a good idea" to "it may be a good idea"*

Done.

*26. Page 11, Lines 9–15: Was this case during precipitation? If so, it should be noted.*

No, this was during clear weather. This piece of information has been added to the manuscript.

*27. Page 11, Lines 21–23: This is a run-on sentence. Add commas where appropriate.*

Done.

*28. Page 12, Lines 3–4: Regarding the final sentence in this section; the improvement of the reflectivity factor data is qualitative at best; no bias estimation is provided, especially in the precipitation regions. You have suppressed turbines in some instances, but it's not appropriate to conclude that you have made a "large improvement" on the data. The subjectivness of this means the sentence should be removed.*

The sentence has been rewritten.

*29. Page 12, Line 24: "amplitudes" should be singular*

Done.

*30. Page 12, Line 33: Remove "significantly" — it is subjective, as we have seen nothing quantitative to suggest a statistically significant removal of clutter, especially in precipitation. The results are simply plots of before and after; they are qualitative in nature*

Done.